# Tractography Alterations in the Arcuate and Uncinate Fasciculi in Post-Stroke Aphasia

**DOI:** 10.3390/brainsci11010053

**Published:** 2021-01-05

**Authors:** Sara Kierońska, Milena Świtońska, Grzegorz Meder, Magdalena Piotrowska, Paweł Sokal

**Affiliations:** 1Department of Neurosurgery and Neurology, Jan Biziel University Hospital No. 2, Collegium Medicum, Nicolaus Copernicus University, 85-168 Bydgoszcz, Poland; sara.kieronska@biziel.pl (S.K.); m.switonska@cm.umk.pl (M.Ś.); piotrowska.magdalena@wp.pl (M.P.); 2Faculty of Health Science, Ludwik Rydygier Collegium Medicum, Nicolaus Copernicus University, 85-067 Bydgoszcz, Poland; 3Department of Interventional Radiology, Jan Biziel University Hospital No. 2, 85-168 Bydgoszcz, Poland; grzegorz.meder@gmail.com

**Keywords:** diffusion-tensor imaging, diffusion-tensor tractography, stroke, arcuate fasciculus, thrombectomy

## Abstract

Fiber tractography based on diffuse tensor imaging (DTI) can reveal three-dimensional white matter connectivity of the human brain. Tractography is a non-invasive method of visualizing cerebral white matter structures in vivo, including neural pathways surrounding the ischemic area. DTI may be useful for elucidating alterations in brain connectivity resulting from neuroplasticity after stroke. We present a case of a male patient who developed significant mixed aphasia following ischemic stroke. The patient had been treated by mechanical thrombectomy followed by an early rehabilitation, in conjunction with transcranial direct current stimulation (tDCS). DTI was used to examine the arcuate fasciculus and uncinate fasciculus upon admission and again at three months post-stroke. Results showed an improvement in the patient’s symptoms of aphasia, which was associated with changes in the volume and numbers of tracts in the uncinate fasciculus and the arcuate fasciculus.

## 1. Introduction

Stroke is the second leading cause of death and one of the most common causes of disability in adults worldwide [1]. Recent studies have shown that 26% of stroke survivors have disabilities related to activities of daily living (ADL). Fifty percent of people who have suffered a stroke have reduced mobility due to hemiparesis [2]. Aphasia is one of the most devastating cognitive impairments following stroke, affecting 21–38% of acute stroke patients [3]. Aphasia following stroke can affect communication and quality of life. About one third of patients report persistent aphasia at least three months following a stroke [4]. Functional recovery most likely involves both spontaneous and learning-dependent processes [5].

Conventional computed tomography (CT) and magnetic resonance imaging (MRI) techniques play an important role in the acute diagnosis and medical management of stroke patients. Lesion characteristics are undoubtedly associated with functional impairments at the time of scanning, and likely predict long-term clinical outcomes [6]. More recent neuroimaging techniques, such diffusion tensor imaging (DTI) and tractography, can be used to aid in the differentiation between those who recover vs. don’t recover. These techniques can also identify patients who will respond vs. not respond to different therapeutic approaches (e.g., rehabilitation) [7].

Magnetic resonance tractography is a valuable neuroimaging type of DTI that allows for the in vivo visualization of white matter pathways. This technique leverages the phenomenon of anisotropic diffusion of water to visualize the axons of nerve cells. In this technique, the diffusion of water protons is determined by coefficients that characterize the direction of diffusion change. The highly organized structure of the axon and natural barriers (e.g., cell membranes, myelin sheaths) determine the anisotropic direction of diffusion [7,8]. DTI parameters, including fractionated anisotropy (FA) and mean diffusivity (MD) parameters, allow for the more precise differentiation of grey and white matter and to track the course of nerve pathways. Given that the values of FA and MD parameters are closely related to the microstructure of neurons, this allows for the quantitative and qualitative assessment of changes occurring in the brain in the course of natural and pathological processes [8]. Importantly, the parameters obtained on the basis of DTI imaging can serve as a kind of marker, helpful in assessing the advancement of a stroke, predicting response to therapies or their long-term results [9,10]. Due to the fact that motor deficits are a particularly common consequence of stroke, the area of interest in research on the use of DTI as a biomarker was in the cortical-spinal system (CST) [9]. It has been shown that there is a significant correlation between the parameters obtained thanks to the DTI technique and the results of physical fitness obtained subsequently. This dependence concerned the FA parameter, which reflects the regenerative abilities of CST, turns out to be particularly useful, making it a potential marker of motor outcomes recovery [10,11]. However, it should be remembered that, in order to unequivocally determine the usefulness, further multi-center studies are necessary, allowing for the standardization of the obtained data [9].

Moreover, important fiber tract, especially in the aspect of post-stroke aphasia is arcuate fasciculus which is the white matter pathway connecting the Broca speech center (located in the frontal lobe) with the Wernicke speech center (temporal lobe). The AF is a bundle of associative fibers that connect Wernicke’s area, which is responsible for sensory aspects of speech, with Broca’s area, which is responsible for motor aspects of speech. This fiber bundle is largely responsible for central speech control by coordinating information between the motor and sensory speech areas [12]. The uncinate fasciculus as a hook fascicle connecting the prefrontal cortex and temporal lobe. The UF mainly interconnects regions that support empathy and emotional responses.

Transcranial stimulation with direct current (tDCS) is a non-invasive method of neurorehabilitation based on neuromodulation. The technique consists of placing electrodes on the head soaked in a physiological saline solution and in contact with a direct current source with an intensity in the range of 1–2 mA. Although the exact mechanism of operation of this technique is still unclear, it is postulated that the current flowing through the cerebral cortex modulates cortical excitation by changing the potential of the cell membrane of neurons and the activity of neuronal receptors [13,14]. The type of modulation depends on the polarity of the electrode located above the area to be treated. The use of anodic stimulation is associated with the intensification of cortical excitation and spontaneous cortical activity, while the use of a cathode has the opposite effect [13,14,15]. The available literature data indicate that tDCS contributes to the improvement of language and behavioral functions in patients with post-stroke aphasia [14,15,16]. It has been suggested that tDCS, when used in the treatment of aphasia, in combination with commonly used neurologopedic rehabilitation techniques is a potentially beneficial therapeutic tool [16].

The aim of the case report is present tractography as a method to assess neural functional rehabilitation after stroke. Moreover, in this study, we explored how neuroimaging, and specifically diffusion tensor tractography, might add to predictive models of stroke functional recovery. To address this, we reviewed key findings in the literature and present data from a relatively rare case wherein DTI findings were important for understanding deficits over time.

## 2. Case

### 2.1. Case History

This case study reports on a 41-year old, right-handed man with no past medical history. The patient was admitted to the Stroke Intervention Treatment Unit in the Department of Neurosurgery and Neurology in University Hospital No. 2 in Bydgoszcz due to difficulty speaking and inability to understand words for five hours. Total sensory and partial motor aphasia was diagnosed, which gave us the diagnosis of mixed aphasia. The patient’s consciousness was clear without evident abnormalities in muscle tone, reflex, or gait. The patient gave informed consent form on the usage of personal data.

### 2.2. Results of Tests

The patient’s vital signs were as follows: temperature: 37 °C orally, BP: 170/90 mmHg, heart rate: 86 beats/min, respiratory rate: 16 breaths/min. The clinical neurological deficits of the patient were assessed using the National Institutes of Health Stroke Scale (NIHSS) and were rated 6 points- Responsiveness 0 point (responsive), Questions 2 points (patient didn’t’ correctly answer either question), Commands 2 points (patient didn’t correctly perform tasks), Horizontal Eye Movement 0 point (normal), Visual field test 0 point (no vision loss), facial palsy 0 point (normal), Motor arm 0 point (no arm drift), Motor leg 0 point (no motor drift), Limb ataxia 0 point (normal), sensory 0 point (no sensor loss), and Speech 2 points (Severe aphasia)

Brain CT scan performed 30 min after admission to the hospital didn’t reveal any abnormalities. However, CT scan performed after 24 h after admission to the hospital showed moderate ischemic infarction in the left temporal lobe (Figure 1). Additionally, 48 h after admission to the hospital, MRI examination was performed (Figure 2). Three months after ischemic stroke, MRI was performed according to the same protocol as baseline (Figure 3.)

### 2.3. Treatment

The patient underwent mechanical thrombectomy within six hours after symptom onset. The patient did not qualify for intravenous thrombolysis due to exceeding the time criterion (i.e., 4.5 h from the onset of stroke symptoms). During digital subtraction angiography (DSA), an occlusion of the inferior trunk of the left middle cerebral artery (MCA) was found. Using a small stent retriever (Catch mini 3 × 15 mm, Balt), mechanical thrombectomy (MT) was performed and a thrombus from the affected branch of the MCA was removed. A control run revealed a short, narrowed segment in the treated MCA branch, which quickly started to thrombose again. Two more passes with the device were performed and yielded the same result, which raised the suspicion of branch dissection. Implantation of the stent into the affected branch was deliberated, but, finally, further attempts to recanalize were terminated. As a result, a slight reperfusion—TICI 1—was obtained.

### 2.4. Outcome after Treatment

The patient was assessed 24 h following admission to the hospital by a speech therapist, using the Frenchay Aphasia Screening Test (FAST Test) and the Boston Naming Test (BNT Test, elaborate by Pachalska and MacQueen) [17,18]. The FAST Test measures Comprehension, Expression, Reading and Writing, whereas the Boston Naming Test measures naming skills. The patient obtained the following results: FAST 0/30 points, BNT 2/60 points. The patient displayed symptoms of mixed, primarily sensory aphasia and showed significantly impaired understanding of speech, including simple and complex commands. The patient also had profound difficulties in naming the items in the illustrations in the course of the Boston Naming Test (BNT), i.e., he could correctly name only two out of 60 items. The patient was diagnosed with alexia and agraphia.

### 2.5. MRI Protocol

Human brains were scanned by MRI (T1, T2-weighed, and DTI with echo planar imaging) using a 20-channel head/neck coil on a single 3.0 T Siemens Magnetom Aera scanner (Erlangen, Germany). We used the following DTI acquisition parameters: slice thickness 5.0 mm; matrix—128 × 128; field of view—240 × 240 mm; repetition time—3500 ms; echo time—83.0 ms. Outcome measures of interest included fractional anisotropy (FA), mean diffusivity (MD) and apparent diffusion coefficient (ADC) DTI is most commonly performed using a single-shot, spin-echo, echo planar image acquisition at b-values similar to those used for conventional DWI (typically b = 1000 s/mm^2^). The scanners of MRI and the fiber tracking protocols at baseline and after three months were the same [19,20].

### 2.6. Fiber Tracking Protocol

Diffusion tensor images were processed using DSI studio software, BSD License. A DTI diffusion scheme was used and a total of 60 diffusion sampling directions were acquired. longer tracts [16]. The b-value was 1000 s/mm^2^. The in-plane resolution was 1.95 mm. The slice thickness was 2 mm. A deterministic fiber tracking was used. A total of 15,000 tracts were calculated. ROIs were defined automatically based on an anatomical atlas loaded into the DSI studio program.

By determining arcuate fasciculus, the first Region of Interest (ROI) was drawn in the coronal section under the central sulcus, and a second ROI had been drawn in the axial view at the temporo-parietal junction. Moreover, for uncinate fasciculus, the first ROI was plotted such that it covered the entire temporal lobe; the second ROI covered the projections over the frontal lobe.

### 2.7. tDCS Protocol

The patient received scheduled, conventional rehabilitation (exercises with a physiotherapist in the rehabilitation room) and received transcranial direct current stimulation (tDCS, Sooma Helsinki, Finland) as a rehabilitation tool during the first 10 days following stroke. TDCS was administered for 30 min daily with 2 mA amplitude. According to protocol which was used in Sebastian’s et al. article, anodal tDCS was applied to the left hemisphere language areas to increase cortical excitability (increasing the threshold of activation), and cathodal tDCS was applied to the right hemisphere homotopic areas to inhibit over activation in contralesional right homologues of language areas [21].

### 2.8. Outcome at Discharge

On the tenth day of hospitalization, the patient underwent a follow-up neurologopedic evaluation. The patient obtained the following results: FAST 20/30 points, BNT 45/60 points. A reduction in aphasic disorders and an improvement in verbal and logical contact with the patient were observed. The patient understood and followed simple and complex commands. Improvements in updating the names of items were observed (perseveration, verbal paraphrases, amnesty gaps). The patient was helped by the hint of the first syllable. The patient correctly recreated the automated word sequences (counting, names of days of the week, months). He was repeating simple sounds and single words correctly. Difficulties with longer words with a complicated grammatical structure and sentences (phonetic paraphrases) were still difficult to repeat. Improvement in reading function was observed in the patient. The patient was correctly recognizing names of individual letters, reading individual words and sentences (paralexia persisted). The patient was properly oriented auto- and allopsychically. The patient was advised to continue further speech therapy after discharge from the hospital ward provided by trained and qualified speech and language therapists, including tasks devoted to naming, comprehension, and increasing verbal output.

The patient received ongoing treatment of ASA 75 mg, enoxaparine 40 mg, and cerebrolysin 30 mg daily starting from the second day of hospitalization. The patient was discharged from the hospital 14 days following stroke and received ASA 150 mg, Atorvastatin 40 mg, Citicoline 1000 mg daily for the next 20 days.

### 2.9. Outcome at Follow Up

Three months after the stroke, the patient underwent a follow-up neurologopedic evaluation. The speech therapy examination upon admission and after three months was performed by the same speech therapist. The patient obtained the following results: FAST 29/30 points, BNT 60/60 points. The assessment showed full independence of the patient, and he was able to perform all social functions, but with slight aphasic disorders. These assessments indicated that the patient would soon be able to start gainful employment, which was important for him to return to pre-stroke functioning and regain lost social roles. Three months after the stroke, the patient was allowed to return to work as a sales representative.

## 3. Analysis

Based on MRI with DTI, the arcuate fasciculus (AF) and uncinate fasciculus (UF) were delineated upon admission (AF Figure 4, UF Figure 5) and three months following the ischemic stroke (AF Figure 6, UF Figure 7). Analyses computed the number of fibers, tract length, and volume. 

Differences in the parameters of the nerve tracts upon admission and three months after hospitalization for the UF and AF are presented in Table 1 and Table 2, respectively.

## 4. Discussion

To the best of our knowledge, this is the first case report on the use of tractography to monitor processes related to neural functional restoration in a patient after thrombectomy. However, the use of the DTI technique to assess the functional recovery of selected areas of the cerebral cortex has been in the past. An example of this is the case by Segihier et al. from 2005 concerning the restoration of visual ability in a patient after a perinatal stroke. The authors of this study also pointed to the usefulness of information obtained thanks to the DTI technique in planning rehabilitation and in predicting the return of functions lost due to brain damage [22,23]. In the present study, we found substantial differences over time in the values of all DTI parameters measured in this study within the AF and UF.

In this study, the patient’s stroke occurred in the left hemisphere, which may have had a greater impact on the UF in the left temporal lobe as compared to the right side [24]. Indeed, we found a decrease in the number of fibers and the volume of UF in within the area affected by the stroke, as compared to the UF in the opposite hemisphere. Perhaps the differences reflect the damage caused by a stroke. However, it is impossible to unequivocally confirm the above thesis because we do not have adequate data from before the onset of a stroke episode. Notably, we observed an increase in the number of fibers and tract volume in both the AF and UF in the left hemisphere, after the application of tDCS and conventional rehabilitation. These parameters were reassessed after three months.

There are a number of changes in the cytoarchitectonics of nervous tissue following ischemic stroke. These neural changes can lead to the loss of axon integrity, which can indirectly result in FA changes. The available data regarding changes in FA in the acute infarction foci are contradictory. Indeed, some studies show reduced FA and others report increased FA [25,26,27]. One theory for these discrepancies was developed by Ozsunar et al. The authors demonstrated that the FA parameter is inversely correlated with signal intensity in the T2-dependent sequence [27]. Therefore, changes in the value of FA in the early stage of the stroke are secondary to oedema, membrane damage, and cell lysis, and may be an indicative measure of the severity of damage. Post-stroke FA in the early stages may therefore be a potential prognostic indicator of infarct recovery [28].

Several studies have used tractography to visualize damage to the corticospinal tract in patients with motor disorders who experience chronic strokes. These studies demonstrate a correlation between the obtained tractography results and the patients’ clinical functioning [29]. Therefore, tractography measures may be a useful prognostic indicator for the return of motor functions following rehabilitation procedures. Indeed, increased FA has been reported following rehabilitation, which may be due to remodeling of white matter motor pathways [30,31]. Tractography can also be used to assess non-motor routes [12,32,33]. Damage to the AF is associated with the occurrence of conductive aphasia, which is characterized by a strongly limited or inability to repeat words heard and a lack of control over active speech [34]. In addition, a study by Breier et al. [35] found that lesions within the AF can cause difficulties not only with repeating heard words, but also with understanding them. These findings point to the relationship between AF damage and changes within posterior superior, middle temporal, and supramarginal gyri that may mediate deficits in speech understanding [35]. A study by Kim and Jang indicates the usefulness of tractography for visualizing the left AF in the early stages of stroke, which suggests that the present approach may be useful for predicting outcomes of aphasia among stroke patients [36]. DTI may also be a valuable diagnostic tool to assess the presence and severity of AF lesions, track the prognosis of aphasia, and identify mechanisms of aphasia regeneration among stroke patients. Together, these applications may aid in the development of a personalized neurologopedic rehabilitation schedule [36,37].

One of the weaknesses of our study is that we didn’t attempt to determine psychological assessment of our patient in terms of empathy and emotional disorders. It was mainly related to the patient’s aphasia. Oishi et al. in their study showed a relationship between reduced volume in the uncinate fasciculus and errors on empathy tasks [38]. Previous studies have also proved a relationship between reduced volume of UF and errors on empathy tasks in some neurological disorder. Results confirm that acute damage to the right UF can disrupt performance on a task of emotional empathy [38,39]. This is the reason that the psychological assessment of stroke patients is so important in future studies, taking into account the aspects of emotions and empathy with correlation with the anatomy of UF. The key goal of clinical management among patients with ischemic stroke is to diagnose stroke as early as possible, preferably during the hyperacute phase, in which reperfusion treatment is the most effective. Identification of patients in the hyperacute phase requires an imaging technique that allows for optimal imaging of the infarct focus [40]. These findings also support further investigation into the use of tDCS as a promising method for promoting speech rehabilitation among ischemic stroke patients. Prior studies by Nitche, Paulus, and Jude suggest that tDCS might prime the brain as an adjuvant to behavioral aphasia and motor-limb therapies, and thus optimize recovery. In particular, a low-intensity current (e.g., 1–2 mA) can be useful for modulating neuronal activity during stroke rehabilitation, which may encourage brain plasticity [41,42].

Thus far, the available literature data indicate a low or moderate quality of evidence regarding the use of tDCS in the treatment of aphasia [13]. Nevertheless, numerous reports suggest that the use of tDCS in the rehabilitation of post-stroke aphasia patients may enhance the process of speech recovery [43,44]. At the same time, despite promising reports in the field of neurorehabilitation using tDCS stimulation, this method still has the experimental status. The tDCS method can be used, among others as non-pharmacological support in the treatment of strokes. There is a need for further research, thus constituting a promising additional tool in recovery [44]. Hesse et al. were the first to pay attention to the changes in neurological status after using of tDCS in patients after stroke. The researchers wanted to check 10 patients with paresis in terms of tolerance of transcranial electro-stimulation. They joined 30 movement training sessions 7-min 1.5 mA a-tDCS, aimed at the motor cortex hemisphere (anode above C3 / C4 acc. To 10/20 electroencephalographic system). The analysis of the results showed that in 4 out of 5 patients with right-sided paresis accompanied by aphasia, language disorders unexpectedly decreased, assessed using the aphasia test (Aachener Aphasie Test) [45].

In an early phase of ischemic stroke, the phenomenon of spontaneous improvement of the neurological state is observed, including remission in terms of aphasia, especially in groups of young patients, which is directly related to brain neuroplasticity [46,47]. To activate the neuroplasticity process, it is important to implement various forms of neurorehabilitation, including tDCS. In the study, Pelegrino et al. proved that neural plasticity is induced by electrical stimulation of tDCS in sensorimotor areas. In this study, tDCS was compared to sham tDCS. The presented analysis shows that bilateral tDCS stimulation increases the number of gamma waves in the brain, which may be related to the neuroplasticity [48,49,50].

In the case of ischemic stroke diagnosis, rapid therapeutic interventions are required due to the short therapeutic window. Successful treatment does not end with reperfusion therapy. The next stage of therapy involves the development of a personalized rehabilitation plan, which aims to reduce neurological deficits as quickly and effectively as possible. The present study demonstrates that tractography may be useful in the diagnosis and monitoring of patients with acute ischemic stroke. Of course, the description of one case is a limitation. Therefore, further studies that include a large group of patients and aim to examine effectiveness are needed. Tractography is a promising new tool for the evaluation of patients following ischemic stroke [51,52].

## 5. Conclusions

This case report demonstrated DTI with tractography as an MRI technique used to detect the microstructural changes and difference in the anatomy and morphology of fiber tracts in patients after ischemic stroke which corresponded with improvement in the patient’s clinical functioning. In the described case, the correlation between the patient’s clinical improvement and neurorehabilitation enhanced by tDCS cannot be clearly defined. Based on the description of one case, no objective conclusions can be drawn regarding the use of post-stroke tractography and tDCS. The description of this case prompts research on a larger group of patients after ischemic stroke.

## Figures and Tables

**Figure 1 brainsci-11-00053-f001:**
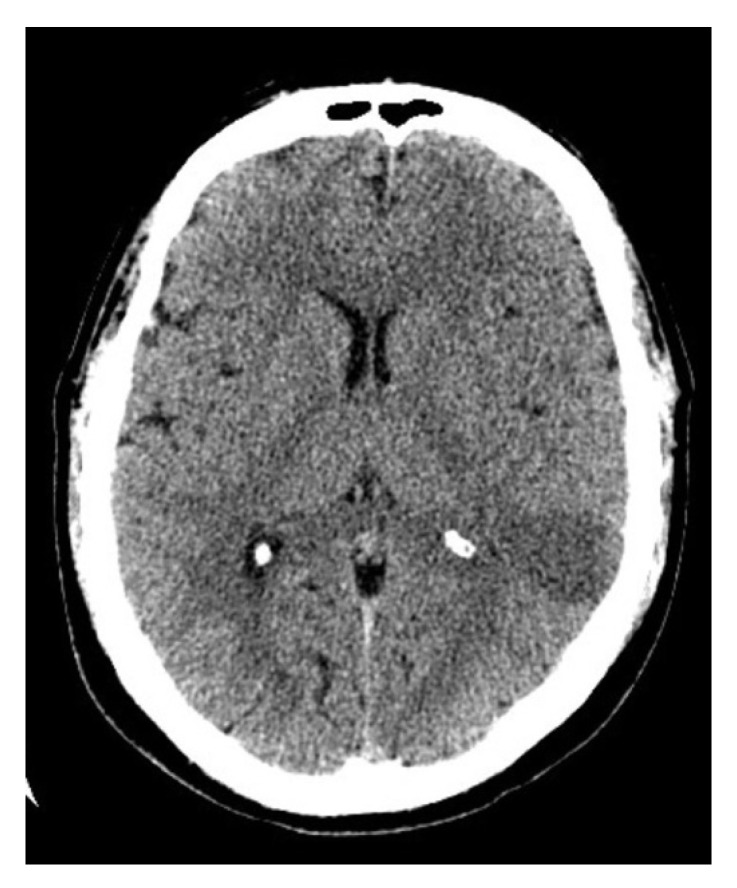
CT scan of the brain after 24 h after admission to the hospital shows an ischemic stroke (30 × 25 × 60 mm) in the left temporal lobe.

**Figure 2 brainsci-11-00053-f002:**
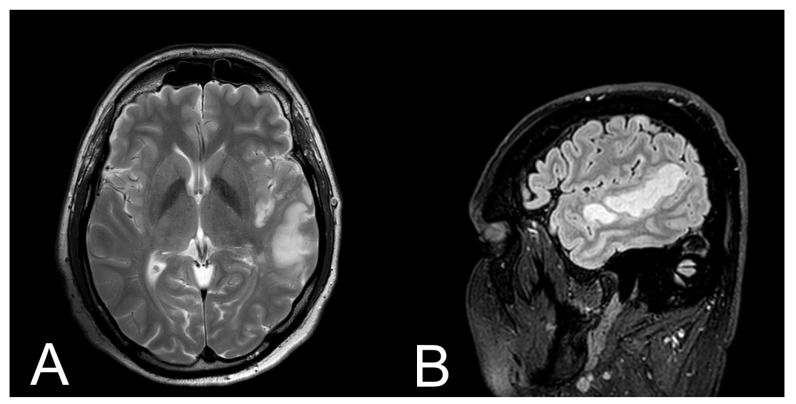
Initial conventional magnetic resonance imaging (MRI) of the patient. (**A**) T2-weighted image with axial plane (T2WI) showed moderate ischemic lesion in the left temporal lobe; (**B**) FLAIR image with sagittal plane showed ischemic lesion with oedema of the left temporal lobe.

**Figure 3 brainsci-11-00053-f003:**
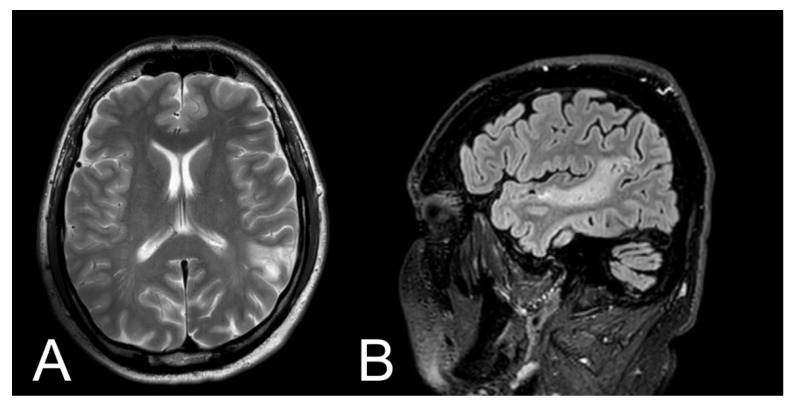
Magnetic resonance imaging (MRI) of the patient after 3 months. (**A**) T2-weighted image with axial plane (T2WI) showed decrease ischemic lesion in the left temporal lobe; (**B**) FLAIR image with sagittal plane showed decrease ischemic lesion without oedema of left temporal lobe.

**Figure 4 brainsci-11-00053-f004:**
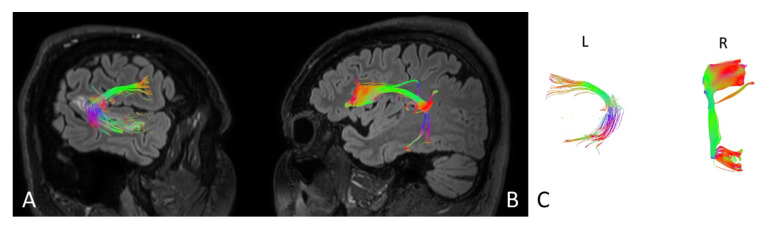
Anatomy of the arcuate fasciculus (AF) upon the patient’s admission to the hospital. (**A**) right hemisphere, (**B**) left hemisphere, (**C**) comparison of left and right axial plane of AF.

**Figure 5 brainsci-11-00053-f005:**
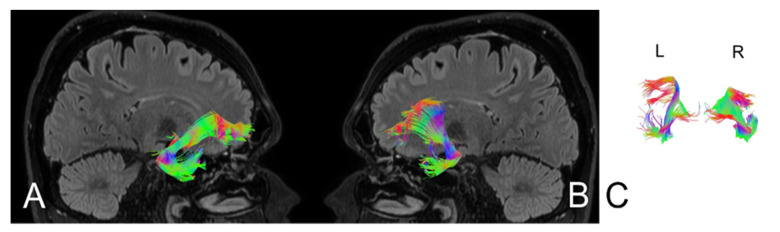
Anatomy of the uncinate fasciculus (UF) upon the patient’s admission to the hospital. (**A**) right hemisphere, (**B**) left hemisphere, (**C**) comparison of left and right axial plane of UF.

**Figure 6 brainsci-11-00053-f006:**
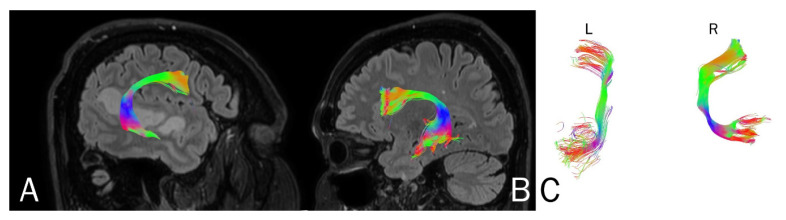
Anatomy of the patient’s arcuate fasciculus (AF) at 3 months following stroke. (**A**) right hemisphere, (**B**) left hemisphere, (**C**) comparison of left and right axial plane of the AF.

**Figure 7 brainsci-11-00053-f007:**
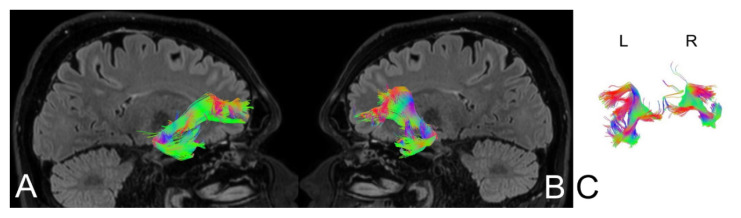
Anatomy of the patient’s uncinate fasciculus (UF) at 3 months following stroke. (**A**) right hemisphere, (**B**) left hemisphere, (**C**) comparison of left and right axial plane of the UF.

**Table 1 brainsci-11-00053-t001:** The patient’s morphological data of the uncinate fasciculus (UF) upon admission and three months following stroke.

	Admission		3 Month Follow-Up	
Parameters	Left side	Right side	Left side	Right side
Number of fibers	148	183	155	152
Volume of tract [mm^2^]	13305	16440	14400	16338
Length of tract [mm]	88.5	98.3	87.5	97.1
FA	0.675	0.895	0.705	0.880

**Table 2 brainsci-11-00053-t002:** The patient’s morphological data of the arcuate fasciculus (AF) upon admission and three months following stroke.

	Admission		3 Month Follow-Up	
Parameters	Left side	Right side	Left side	Right side
Number of fibres	160	205	185	208
Volume of tract [mm^2^]	1125	18440	1280	18559
Length of tract [mm]	75.5	88.0	86.4	89.0
FA	0.715	0.885	0.785	0.880

## Data Availability

The datasets generated during and/or analysed during the current study are available from the corresponding author on reasonable request.

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
