# Peer review of "Tractography Alterations in the Arcuate and Uncinate Fasciculi in Post-Stroke Aphasia"

_brainsci, 2021, doi:10.3390/brainsci11010053_

Round 1
Reviewer 1 Report
Kieronska et al. presented a case report on tractography and language testing collected from a single patient immediately and three months after a left MCA stroke. The primary weakness of the study is that data from a case report methodology cannot support the claims stated in the conclusion. Furthermore, as a case report, the study is not novel. Many studies for example on the correlations between arcuate fasciculus tractography and language performance post stroke have been published.
Methodology of the neuroimaging acquisition and analysis, which is the central component of this report, is severely lacking. The description of the tDCS protocol could also be improved, e.g. the device used. In my opinion, majority of the current materials and methods can more succinctly summarized. Also, there should be a better delineation of the methods and results sections. It's also unclear whether any changes were observed in the uncinate fasciulus. The study might be strengthened by 1) additional patients, and 2) modification in study design to include for instance control groups or off/on data points within fewer patients.
Author Response
Answer review 1
Thank you very much for a very insightful and valuable review of our article. We hope that thanks to the comments we were able to improve the content of the article so that it met expectations and could be published.
In our centre, we have just started performing tractography in patients with ischemic cerebral incidents. Our work is a pilot study, so we treat the description of the patient as a case report. We intend to prepare an original paper with more cases of ischemic stroke treated with tDCS for whom treatment will be scheduled. However, due to the short time of using tDCS and tractography in our centre, we currently describe only one case. Also, we wanted to present our tractography protocol and presentation of nerve fibres using the DSI studio program, which is new in our centre.
As suggested by the reviewer, the methodology was improved with the extension of the parameters of the tDCS used: "The patient received scheduled, conventional rehabilitation, and received transcranial direct current stimulation (tDCS (Sooma Ltd) Helsinki, Finland) as a rehabilitation tool during the first 10 days following stroke. TDCS was administered for 30 minutes daily with 2mA amplitude. According to a protocol which was used in Sebastian's et al article anodal tDCS was applied to the left hemisphere language areas to increase cortical excitability (increase the threshold of activation) and cathodal tDCS was applied to the right hemisphere homotopic areas to inhibit over-activation in contralesional right homologues of language areas”.
Besides, as suggested by the reviewer, a detailed MRI image acquisition protocol and basic tractography parameters have been added: Human brains were scanned by MRI (T1, T2-weighed, and DTI with echo-planar imaging) using a 20-channel head/neck coil on a single 3.0 T Siemens Magnetom Aera scanner (Erlangen, Germany). We used the following DTI acquisition parameters: slice thickness 5.0 mm; matrix—128 x 128; a field of view—240 x 240 mm; repetition time—3500 ms; echo time—83.0 ms. Outcome measures of interest included fractional anisotropy (FA), mean diffusivity (MD), and apparent diffusion coecient (ADC). DTI is most commonly performed using a single-shot, spin-echo, echo-planar image acquisition at b-values similar to those used for conventional DWI (typically b = 1000 s/mm2). ROIs were defined automatically by the DSI Studio based on an anatomical atlas loaded into it.
Diffusion tensor images were processed using DSI studio software, BSD License. A DTI diffusion scheme was used and a total of 60 diffusion sampling directions were acquired. longer tracts [16]. The b-value was 1000 s/mm2. The in-plane resolution was 1.95 mm. The slice thickness was 2 mm. Deterministic fibre tracking was used. A total of 15,000 tracts were calculated.”
The work has been corrected by a native speaker, we add a translation certificate attached
With best regards
The authors
Reviewer 2 Report
In this study, Kieronska and colleagues present a novel imaging tool using tractography to evaluate patients after an ischemic stroke. Their findings are in general well presented. Language and grammatical editing are highly recommended. The authors should also avoid contractions for formal scientific writing. The methodology and conclusions of this report seem appropriate.
I have very punctual comments and recommendations:
-Methodology section: for the patient’s arrival at the hospital, please clarify the exact onset and timing of the global aphasia, since a Glasgow Coma Scale of 15 points looks confusing for a patient with motor and speech problems.
-Same section: When was exactly the initial MRI performed? How many hours after the insult?
-Results section: How were the fibers of the AF and UF delineated? Please elaborate on the technique. Manually or using an atlas…?
-Same section: Provide specifications about the software used for the tractographies.
Author Response
Thank you very much for an extremely interesting and valuable review.
Taking into account the reviewer's suggestion, the article has been corrected by introducing these corrections and comments:
- The GCS scale is a scale for assessing awareness and not for assessing the severity of aphasia. The patient described in our article was rated 15 points on the GCS-Eyes opening was spontaneously scale, the patient's orientation was full despite the features of aphasia. He complied with all the instructions and showed no signs of paresis. Based on his neurological status, he was rated at 15 GCS points. In our opinion, it is possible despite aphasia. Moreover, the patient described in the article was fully oriented and, despite his aphasia, he communicated by writing words on a piece of paper, hence the patient's assessment of 15 points on the GCS scale.
- Baseline MRI was performed 12 hours after the onset of symptoms
- The article adds data on the protocol used for MRI and information on the deletion of fibres based on the automatic anatomical atlas: “Human brains were scanned by MRI (T1, T2-weighed, and DTI with echo-planar imaging) using a 20-channel head/neck coil on a single 3.0 T Siemens Magnetom Aera scanner (Erlangen, Germany). We used the following DTI acquisition parameters: slice thickness 5.0 mm; matrix — 128 x 128; a field of view — 240 x 240 mm; repetition time — 3500 ms; echo time — 83.0 ms. Outcome measures of interest included fractional anisotropy (FA), mean diffusivity (MD), and apparent diffusion coefficient (ADC). DTI is most commonly performed using a single-shot, spin-echo, echo-planar image acquisition at b-values similar to those used for conventional DWI (typically b = 1000 s / mm2). ROIs were defined automatically based on an anatomical atlas loaded into the DSI studio program. "
- Fiber tracking protocol was added to the article: ”Diffusion tensor images were processed using DSI studio software, BSD License. A DTI diffusion scheme was used and a total of 60 diffusion sampling directions were acquired. longer tracts [16]. The b-value was 1000 s/mm2. The in-plane resolution was 1.95 mm. The slice thickness was 2 mm. Deterministic fibre tracking was used. A total of 15,000 tracts were calculated.”
We hope that the introduced changes will significantly increase the substantive value of the article and meet all the requirements for publication.
With best regards
The authors
Reviewer 3 Report
This topic is quite attractive and meaningful for further investigation of the tDCS. The method part should be improved to present the technical parameters of the fiber tracking process.
Minor points:
1.The part of the clinical estimations is in the method part, while the others in the result part. Could the author re-organize them, and I would suggest using a table to present them?
2. The structure of the article is not ideal for a case report, could the author change the case report to the standard structure suggested by the guideline? (https://www.ncbi.nlm.nih.gov/pmc/articles/PMC5686928/)
3. Only the MRI anatomical images (fig 1) during admission are presented. Here also need to show the MRI images three months after the stroke to show the changes in the brain.
4. It should be careful when interpreting the results, the effect of medicine should be considered. The white matter can be affected by the medicine, too.
Sugimoto K, Kakeda S, Watanabe K, Katsuki A, Ueda I, Igata N, Igata R, Abe O, Yoshimura R, Korogi Y. Relationship between white matter integrity and serum inflammatory cytokine levels in drug-naive patients with major depressive disorder: diffusion tensor imaging study using tract-based spatial statistics. Transl Psychiatry. 2018 Aug 1;8(1):141. DOI: 10.1038/s41398-018-0174-y. PMID: 30069019; PMCID: PMC6070558.
5. In fig 2 A, the tracked figure is described as AF, but the shape of the fiber is different than the AF presented in fig 2B and fig4. Could the author recheck it?
Major points:
1The DTI data is sensitive to the scanning setups. The scanners used in admission and at three months post-stroke are the same one? And their set-ups are the same?
Vollmar C, O'Muircheartaigh J, Barker GJ, Symms MR, Thompson P, Kumari V, Duncan JS, Richardson MP, Koepp MJ. Identical, but not the same: intra-site and inter-site reproducibility of fractional anisotropy measures on two 3.0T scanners. Neuroimage. 2010 Jul 15;51(4):1384-94. DOI: 10.1016/j.neuroimage.2010.03.046. Epub 2010 Mar 23. PMID: 20338248; PMCID: PMC3163823.
Andica C, Kamagata K, Hayashi T, Hagiwara A, Uchida W, Saito Y, Kamiya K, Fujita S, Akashi T, Wada A, Abe M, Kusahara H, Hori M, Aoki S. Scan-rescan and inter-vendor reproducibility of neurite orientation dispersion and density imaging metrics. Neuroradiology. 2020 Apr;62(4):483-494. DOI: 10.1007/s00234-019-02350-6. Epub 2019 Dec 27. PMID: 31883043; PMCID: PMC7093343.
2. To compare the fiber volumes and fiber numbers between the two time-points (admission and at three months post-stroke), the effects caused by the changes between the DTI scannings should be excluded to guarantee the fiber changes is because of the improvement but not by the difference between scan and rescan. Could the author offer some data to support it?
3. Regarding the fiber tracking process, it has not been mentioned in the method part and only one sentence in the results part "Based on MRI with DTI, the arcuate fasciculus (AF) and uncinate fasciculus (UF) was delineated upon admission." Could the author please offer detailed information on the DTI fiber tracking? I would like to suggest to describe the following information, including the software applied for fiber tracking, the algorithms applied to the fiber tracking (deterministic or probabilistic?), the parameters and setups for the fiber tracking, the software used for the screenshot and figures presented in the article, whether the region of interest (ROI) is used or not during fiber tracking, how and where the ROI is set, the ROI is set artificially or according to an anatomical template, Whether the DTI data is registered to an anatomical template?
Author Response
Answer to the review report 3
Thank you very much for the professional analysis and review of the article and all valuable suggestions and comments. I hope that the amendments we made will significantly improve the quality of the article.
As suggested, appropriate changes were made:
- the structure of the article was changed according to the rules of creating a case report
- figure imaging MRI scan in an axial and sagittal plane after 3 months after ischemic stroke and baseline CT examination was added
-Information was added: “The scanners of MRI and the fiber tracking protocols at baseline and after 3 months was the same.” All MRI examinations were performed on the same apparatus with the same scanning parameters. In the case of DTI, the same acquisition parameters and ROI were used for the baseline and 3 months follow up.
In our hospital that performs a large number of MRI examinations with the DTI protocol, each tractography analysis is based on the same acquisition parameters. There is a risk of errors in each analysis. It should be noted that the accuracy of imaging using DTI depends on the homogeneity of the magnetic field gradients, which translate into the value of the b-matrix. System errors, including equipment-dependent restrictions and scanner or gradient coils, are associated with the spatial heterogeneity of diffusion gradients. The spatial heterogeneity of diffusion gradients, in turn, causes vector distortion and ultimately, falsification of the resulting image. Thus, it became necessary to develop a model that would eliminate the problem as much as possible. The first experimental mathematical scheme to retrospectively correct the effects of spatial gradient field distortions on diffusion-dependent imaging was developed by Bammer. However, to optimally eliminate and correct measurements of the diffusion tensor caused by heterogeneity in the magnetic field gradients, the calibration technique BSD-DTI (b-matrix spatial Distribution DTI technique) was developed and introduced, which significantly improved the fiber tracking procedure. In our centre, the BSD-DTI protocol is still under development, therefore we use a deterministic basic protocol.
- Information regarding the determination of ROI was added to the article: "By determined arcuate fasciculus first Region of Interest (ROI) was drawn in the coronal section under the central sulcus, a second ROI had been drawn in the axial view at the temporo-parietal junction. Moreover, for uncinate fasciculus the first ROI was plotted such that it covered the entire temporal lobe; the second ROI covered the projections over the frontal lobe. "
- Description of MR scanning parameters, DTI determination and the tDCS execution protocol has been added to the article: ”Human brains were scanned by MRI (T1, T2-weighed, and DTI with echo planar imaging) using a 20-channel head/neck coil on a single 3.0 T Siemens Magnetom Aera scanner (Erlangen, Germany). We used the following DTI acquisition parameters: slice thickness 5.0 mm; matrix—128 x 128; field of view—240 x 240 mm; repetition time—3500 ms; echo time—83.0 ms. Outcome measures of interest included fractional anisotropy (FA), mean diffusivity (MD), and apparent diffusion coefficient (ADC). DTI is most commonly performed using a single-shot, spin-echo, echo planar image acquisition at b-values similar to those used for conventional DWI (typically b = 1000 s/mm2). ROIs were defined automatically based on an anatomical atlas loaded into the DSI studio program.
Diffusion tensor images were processed using DSI studio software, BSD License. A DTI diffusion scheme was used and a total of 60 diffusion sampling directions were acquired. longer tracts. The b-value was 1000 s/mm2. The in-plane resolution was 1.95 mm. The slice thickness was 2 mm. A deterministic fiber tracking was used. A total of 15,000 tracts were calculated.
- a bibliography suggested by the reviewer was also added to the article.
Sincerely
The authors
Reviewer 4 Report
Dear Authors,
I have had the pleasure of reviewing your article “Tractography alterations in the arcuate and uncinate 2 fasciculi in post-stroke aphasia modulated by tDCS.”
In the article, the authors attempt to link supposed changes in DTI fiber tract parameters induced by tDCS to a rehabilitation of speech capabilities in an individual suffering from stroke.
There is a potential story to tell here. The manuscript in its current iteration however fails to convince me. There are several reasons for this:
Introduction:
- The introduction gives an overview regarding the molecular mechanisms underlying DTI. The section neglects however much of the literature that is already present regarding clinical relevance of DTI in stroke. For a review, see doi: 10.3389/fneur.2019.00445 . I would encourage the authors to add respective introductory segments and references to their article.
- The introduction does not mention tDCS, even though the title would imply a central role of this technology in the case. I think the manuscript would be improved by including a section on tDCS in the introduction explaining the motivation underlying its use in this case.
- Similarly, the arcuate and uncinate fascicles should be introduced with corresponding literature on their respective function.
- Lines 53-55: “Given that the values of FA and MD parameters are closely related to the microstructure of neurons, this allows for the quantitative and qualitative assessment of changes occurring in the brain in the course of natural and pathological processes [8]“ The cited source (doi:10.1159/000442303) does not support this statement. The reference does not mention anisotropy or mean diffusivity. I would urge the authors to be more considerate in their bibliography and cite appropriate sources.
Materials and Methods:
- Lines 63-67: “The patient was admitted to the Stroke Intervention Treatment Unit in the Department of Neurosurgery and Neurology in University Hospital No 2 in Bydgoszcz with global aphasia lasting one hour. The patient’s consciousness was clear without evident abnormalities in muscle tone, reflex, or gait and Glasgow coma scale (GCS) was 15/15.” A global aphasia is per definition not reconcilable with GCS 15/15.
- Lines 69-70: “(…)the National Institutes of Health Stroke Scale (NIHSS) and were rated 6 points.” I would recommend that the authors specify which exact subcategories were rated in what way to achieve 6 points.
- In general, the initial case presentation does not adequately present the time course of the case. I would recommend that the authors either create a figure or modify the text to precisely describe the exact time course starting from symptom onset, going over arrival at the hospital, individual imaging modalities with respective findings, thrombectomy et cetera. This I believe is necessary in order to let the reader develop a proper understanding of the clinical case.
- The manuscript does not mention any form of cranial CT imaging which is the usual standard for initial stroke diagnostics. If cranial CT was obtained before DSA, I would suggest that the result be described in the article.
- The linguistic tests used should be referenced properly including distributor, city, country etc.
- Lines 86-87: “Due to impaired auditory control, the patient was unable to control what he said.” This sentence is not understandable, at least to me. Does it mean to indicate that the patient was unable to speak due to hearing problems? It should be rephrased more precisely.
- Lines 87-88: “The patient also had profound deficits in updating the names of the items in the illustrations.” It should be clarified in the manuscript which task in which test is meant with reference to the illustrations.
- Lines 89-90: “Initial head MRI with a DTI protocol scan showed moderate ischemic infarction of the left temporal lobe (Fig. 1).” Fig. 1 does not show a DTI protocol. The authors may want to rephrase the sentence.
Results:
- Lines 100-103: “The patient received scheduled, conventional rehabilitation, and received transcranial direct current stimulation (tDCS) as a rehabilitation tool during the first 10 days following stroke. TDCS was administered for 30 minutes daily, using anodal stimulation of left frontal lobe (ampitude 2mA).” Firstly, the conventional rehabilitation should be detailed more thoroughly. What does it consist of? Secondly, since tDCS was a non-factor in the introduction, there is a lack of explanation regarding the reasoning behind the use of the stimulation. Why is anodal stimulation of left frontal lobe used? An adequate section in the introduction including relevant literature could help with this.
- The patient’s clinical state at discharge should be detailed.
- Lines 106-110: “Based on MRI with DTI, the arcuate fasciculus (AF) and uncinate fasciculus (UF) were delineated upon admission (AF Fig. 2, UF Fig. 3) and three months following the ischemic stroke (AF Fig. 4, UF Fig. 5). Analyses computed the number of fibers, tract length, and volume. The MRI examination upon admission and after three months was performed on the same 3.0 T Siemens Magnetom Area MRI scanner (Erlangen, Germany) equipped with a 20-channel head/neck coil.” This segment is severely lacking in methodological detail. Which program was used for tractography? What were the detailed parameters of the DTI sequence during MRI imaging (e.g. number of directions)? Was the tractography performed according to deterministic or probabilistic fibertracking? Was the tracking based on ROIs, and if so, which were used? Which variables were fixed input, and which were changed to optimize tracking results? DTI tractography is highly sensitive to small changes in tracking parameters. The present iteration of the manuscript does not allow for any form of replication of the tractography procedure. This is one of the most significant problems with the current manuscript.
- Line 117 indicates the patient had residual aphasia. The nature of the residual symptoms should be explained more clearly.
- Due to the lack of methodological detail, it is not possible to gauge the value of the results presented in Figures 2-5 and Tables 1-2.
Discussion:
- Lines 146-147: “To the best of our knowledge, this is the first case report on the use of tractography to monitor processes related to neural plasticity in a patient undergoing thrombectomy.” While potentially true regarding the factor of thrombectomy, rather similar cases have already been reported as far back as 2005, e.g. https://doi.org/10.1186/1471-2377-5-17 . The authors might consider referencing cases such as this one.
- Lines 151-153: “Indeed, we found a decrease in the number of fibers and the volume of UF in within the area affected by the stroke, as compared to the UF in the opposite hemisphere.” In my opinion, this statement holds little scientific value within the context of this case, since no comparison to the patient’s pre-stroke fiber architecture can be made. I.e., what if there was an imbalance in the number of fibers and volume even before the stroke?
- Lines 156-157: “After three months, lower FA in the AF positively correlated with greater deficits in speech understanding.” It is unclear to me what the authors try to communicate here. The general understanding at this point in the article is that the patient had better speech capabilities after three months. This sentence seems to suggest otherwise?
- Line 193: “One advantage of tractography is the possibility of visualizing white matter pathways covered by the stroke foci at an early stage of its occurrence [23]” The article cited at the end (doi:10.1016/j.jstrokecerebrovasdis.2018.02.043.) does not directly support the prior statement.
- The role of the uncincate fascicle is hardly discussed. This should be changed.
Conclusions:
- Lines 213-216: “This case report demonstrated that properly applied immediate neurorehabilitation augmented by neuromodulation (i.e., tDCS), even after failed endovascular treatment, can improve the patient's neurological condition. Results of the present study suggest that tDCS-enhanced 215 rehabilitation may be effective in the management of young stroke patients.“ In my opinion, this statement is not validated in its totality. The authors imply that neuromodulation via tDCS had a role in the patient’s recovery, yet there is no direct evidence in this case report that supports this hypothesis. This is a fundamental limitation of a one-person intervention study. Without a control group, in my professional opinion, no statement regarding the individual effect of neuromodulation on the patient’s recovery can and should be made. I would therefore ask the authors to adopt a more careful interpretation of their results.
General Comments:
- The article is not structured as would be expected for a case report. Regarding subsections of the methods and results sections, I would urge the authors to structure their article with subheadings similar to reports such as https://doi.org/10.3390/brainsci10110883 or https://doi.org/10.3390/brainsci10100718 This would improve the general readability.
- There seems to be no mention of the patient consenting to the use of his data. Informed consent should be obtained, or if it has been obtained already it should be specified.
- To me, the article lacks a clear direction. It does not discuss tDCS in a significant manner, it does not go deep enough into DTI tractography, it does not attribute new specific function to the respective fiber tracts. In this iteration, the manuscript does not seem to have a clear purpose that contributes new perspectives to the existing literature.
- The manuscript’s language could be improved via proofreading by a native speaker.
In conclusion, I believe that the article currently has many weaknesses for publication.
I do however believe that the author’s tDCS/DTI tractography approach is potentially interesting and I encourage them to pursue e.g. a controlled study to clearly identify the potential benefit of tDCS in stroke rehabilitation as related to tractography changes.
Author Response
Answer for the review report 4
Thank you very much for the professional and very thorough analysis and review of the article and all valuable suggestions and comments. It was a pleasure for the authors to introduce changes to the article after such a professional analysis. I hope that the amendments made will significantly improve the quality of work.
According to reviewer's suggestions, the following changes were made:
- MRI scan figure of the axial and sagittal plane after 3 months after ischemic stroke and baseline CT examination
-References to linguistic tests had been added
-The protocol of usage of tDCS with a description of application of electrodes and relevant literature has been added: ”The patient received scheduled, conventional rehabilitation, and received transcranial direct current stimulation (tDCS (Sooma Ltd) Helsinki, Finland Licence) as a rehabilitation tool during the first 10 days following stroke. TDCS was administered for 30 minutes daily with 2mA amplitude. According to protocol which was used in Sebastian’s et al article anodal tDCS was applied to the left hemisphere language areas to increase cortical excitability (increase the threshold of activation) and cathodal tDCS was applied to the right hemisphere homotopic areas to inhibit over activation in contralesional right homologues of language areas. “
-According to the suggestion, the detailed description of patient’s condition during the discharge has been added: „On the tenth day of hospitalization, the patient underwent a follow-up neurologopedic evaluation. The patient obtained the following results: FAST 20/30 points, BNT 45/60 points. A reduction in aphasic disorders and an improvement in verbal and logical contact with the patient were observed. The patient understood and followed simple and complex commands. Improvements in updating the names of items were observed (perseveration, verbal paraphases, amnesty gaps). The patient was helped by the hint of the first syllable. The patient correctly recreated the automated word sequences (counting, names of days of the week, months). He was repeating sounds, simple, single words correctly. Difficulties, longer words with a complicated grammatical structure and sentences (phonetic paraphases) were still difficult to repeat. Improvement in reading function was observed in the patient. The patient correctly recognized names, individual letters, read individual words, sentences (paralexia persist). The patient was properly oriented auto - and allopsychically. The patient was advised to continue further neurological therapy after discharge from the hospital ward.”
-NIHSS score with subcategories was specified
-Figure of CT scan at baseline with ischemic area in the left temporal lobe was added
-Sentence:” Initial head MRI with a DTI protocol scan showed moderate ischemic infarction of the left temporal lobe “ was changed.
- MRI and fiber tracking protocol parameters were added: " MRI protocol
Human brains were scanned by MRI (T1, T2-weighed, and DTI with echo planar imaging) using a 20-channel head/neck coil on a single 3.0 T Siemens Magnetom Aera scanner (Erlangen, Germany). We used the following DTI acquisition parameters: slice thickness 5.0 mm; matrix—128 x 128; field of view—240 x 240 mm; repetition time—3500 ms; echo time—83.0 ms. Outcome measures of interest included fractional anisotropy (FA), mean difusivity (MD), and apparent diusion coecient (ADC). DTI is most commonly performed using a single-shot, spin-echo, echo planar image acquisition at b-values similar to those used for conventional DWI (typically b = 1000 s/mm2). The scanners of MRI and the fiber tracking protocols at baseline and after 3 months was the same.
Fiber tracking protocol
Diffusion tensor images were processed using DSI studio software, BSD License. A DTI diffusion scheme was used and a total of 60 diffusion sampling directions were acquired. longer tracts [16]. The b-value was 1000 s/mm2. The in-plane resolution was 1.95 mm. The slice thickness was 2 mm. A deterministic fiber tracking was used. A total of 15,000 tracts were calculated. ROIs were defined automatically based on an anatomical atlas loaded into the DSI studio program.
By determined arcuate fasciculus first Region of Interest (ROI) was drawn in the coronal section under the central sulcus, a second ROI had been drawn in the axial view at the temporo-parietal junction. Moreover, for uncinate fasciculus the first ROI was plotted such that it covered the entire temporal lobe; the second ROI covered the projections over the frontal lobe.”
-As the authors of the study, we realize that drawing conclusions about the effectiveness of the tDCS method in terms of improvement of the clinical condition and the number and volume of nerve fibers based on a single patient is controversial. In our centre, we have just started performing tractography in patients with ischemic. Our work is a pilot study, so we treat the description of the patient as a case report, not a case series. We plan to prepare an original paper with more cases of ischemic stroke treated with tDCS for whom treatment will be scheduled. However, due to the short time of using tDCS and tractography in our centre, we currently describe only one case. Also, we wanted to present our tractography protocol and presentation of nerve fibres using the DSI studio program, which is new in our centre.
-Information about patients agreement on personal data usage has been added to the article :” The patient has given inform consent form on the usage of personal data.”
-The article has been translated by the native speaker. Authors can provide the certificate that can confirm this.
-Conclusions in the article were changed to a more careful ones: „Currently, tDCS can be seen to have great potential, in particular in the aphasia after ischemic stroke. The tDCS method can be used, among others, as a non-pharmacological treatment aid in strokes. The tDCS method can be used, among others. as non-pharmacological support in the treatment of strokes. At the same time, despite promising reports in the field of neurorehabilitation using tDC stimulation, this method still has the status of experimental activities”
- The GCS scale is a scale for assessing awareness and not for assessing the severity of aphasia. The patient described in our article was rated 15 points on the GCS-Eyes opening was spontaneously scale, the patient's orientation was full despite the features of aphasia. He complied with all the instructions, and showed no signs of paresis. Based on his neurological status, he was rated at 15 GCS points. In our opinion, it is possible despite aphasia. Moreover, the patient described in the article was fully oriented.
-Moreover, in our hospital that performs a large number of MRI examinations with the DTI protocol, each tractography analysis is based on the same acquisition parameters. There is a risk of errors in each analysis. It should be noted that the accuracy of imaging using DTI depends on the homogeneity of the magnetic field gradients, which translate into the value of the b-matrix. System errors, including equipment-dependent restrictions and scanner or gradient coils, are associated with the spatial heterogeneity of diffusion gradients. The spatial heterogeneity of diffusion gradients, in turn, causes vector distortion and ultimately, falsification of the resulting image. Thus, it became necessary to develop a model that would eliminate the problem as much as possible. The first experimental mathematical scheme to retrospectively correct the effects of spatial gradient field distortions on diffusion-dependent imaging was developed by Bammer. However, to optimally eliminate and correct measurements of the diffusion tensor caused by heterogeneity in the magnetic field gradients, the calibration technique BSD-DTI (b-matrix spatial Distribution DTI technique) was developed and introduced, which significantly improved the fiber tracking procedure. In our center, the BSD-DTI protocol is still under development, therefore we use a deterministic basic protocol.
-Suggested literature has been added
-Moreover, the discussion and conclusions were reorganized: Thus far, the available literature data indicate a low or moderate quality of evidence regarding the use of tDCS in the treatment of aphasia [12]. Nevertheless, numerous reports suggest that the use of tDCS in the rehabilitation of post-stroke aphasia patients may enhance the process of speech recovery [38,39]. It seems that in the case of the patient described in this study, the use of tDCS together with standard techniques of neurologopedic rehabilitation contributed to the recovery of language function. There is a need for further research, thus constituting a promising additional tool in recovery.[39]
Hesse et al. were the first to pay attention to the changes in neurological status after using of tDCS in patients after stroke. The researchers wanted to check 10 patients with paresis in terms of tolerance of transcranial electro -stimulation. They joined 30 movement training sessions 7-minute 1.5 mA a-tDCS, aimed at the cortex movable hemisphere (anode above C3 / C4 acc. To 10/20 electroencephalographic system). The analysis of the results showed that in 4 out of 5 patients with right-sided paresis accompanied by aphasia, language disorders unexpectedly decreased, assessed using the aphasia test (Aachener Aphasie Test)[40].
- The GCS scale is a scale for assessing awareness and not for assessing the severity of aphasia. The patient described in our article was rated 15 points on the GCS-Eyes opening was spontaneously scale, the patient orientation was full despite the features of aphasia. He complied with all the instructions and showed no signs of paresis. Based on his neurological status, he was rated at 15 GCS points. In our opinion, it is possible despite aphasia. Moreover, the patient described in the article was fully oriented and, despite his aphasia, he communicated by writing words on a piece of paper, hence the patient assessment of 15 points on the GCS scale.
We hope that the introduced corrections will significantly improve the quality of the article, in line with the reviewer's expectations.
With best regards
The authors
Round 2
Reviewer 1 Report
While the authors have provided more details on the methodology, I still believe data from a single patient cannot answer the research question I presume is the rationale of this study: both the clinical value of tractography to assess neuroplasticity and the value of tDCS to contribute to faster functional recovery after stroke. I believe more data, including a study design with control patients or controls in an on/off design, would be more appropriate and justify the value of this publication. It is not possible to discern whether functional or DTI changes are associated with the application of tDCS.
Author Response
We would like to thank the reviewer for this valuable opinion which in general is concordant with our perception. We agree that larger sample of examined patients would stronger demonstrate any relationship between tDCS and neuroplasticity expressed in shape and size of arcuate and uncinate fasciculi. However, our case report is devoted to issue of recovery after stroke and observed changes in fasciculi in the patient who was treated in a routine way, mainly by logopedic neurorehabilitation enhanced only by tDCS. The main idea was to visualize evolution of changes in connectivity associated with the healing and recovery but not to demonstrate effectiveness of tDCS. Therefore, we highlighted in text that we do not associate DTI changes with the application of tDCS.
Thank You once again
Reviewer 3 Report
The authors have changed the case report accordingly, and give a timely reply. The case report is substantially and structurally improved.
Here I would like to give some minor points for revision:
1) I would like to suggest the authors change the "3. results" to the "2.5 Analysis".
2) The Arcuate Fasciculus shape in Figure 3 A is confusing, could authors offer more figures from different views for the screenshots to confirm it, or redo the fiber tracking of the Arcuate Fasciculus? It is a key point in the current case report.
3) It is very encouraging to see the effective tDCS treatment in clinical practice. The focus of the case report should be that tDCS contributes to the recovery of the neural function. The involvement of neuroplasticity in this rehabilitation process should be described in the discussion part if the authors want to mention it in the current case report.
For example:
Please rephrase "To the best of our knowledge, this is the first case report on the use of tractography to monitor processes related to neural plasticity in a patient undergoing thrombectomy." to "To the best of our knowledge, this is the first case report on the use of tractography to monitor processes related to neural functional restoration in a patient after thrombectomy.
Please rephrase "The aim of the article is present tractography as a method to assess neuroplasticity after stroke. " to " The aim of the case report is present tractography as a method to assess neural functional rehabilitation after stroke."
4) The two sessions of DTI can be used to demonstrate their corresponding functional tests, while they should not be compared directly to conclude. Because there were no controls in the design to avoid scanning related errors between different scanning phases. This shortcoming needed to be mentioned in the corresponding paragraphs in the report. For example, the 2nd paragraph in the Discussion part.
Author Response
Thank you very much for the professional review on the article. As suggested by the reviewer, the following changes were made:
1) Paragraph numbering has been changed as suggested
2) AF at baseline on the left side has been re-visualized and the graphics have been improved. The program does not offer the possibility of presenting the road in 3D, hence the image was presented in the sagittal projection.
3) As suggested, a paragraph has been added regarding neuroplasticity, rehabilitation and tDCS: "In an early phase of ischemic stroke, the phenomenon of spontaneous improvement of the neurological state is observed, including remission in terms of aphasia, especially in groups of young patients, which is directly related to brain neuroplasticity. [46.47] To activate the neuroplasticity process, it is important to implement various forms of neurorehabilitation, including tDCS. In the study, Pelegrino et al. authors proved that neural plasticity is induced by electrical stimulation of tDCS in sensorimotor areas. In this study, tDCS was compared to sham tDCS. The presented analysis shows that bilateral tDCS stimulation increases the number of gamma waves in the brain, which may be related to the neuroplasticity. [48–50]. Besides, the opinion on the aim of the study has been changed.
4) DTI at baseline and 3 months follow up analyzes were prepared based on identical tractography protocols. Moreover, MRI examinations were also performed on the same diagnostic devices. The DSI studio program that was used to determine the tractography performs data analysis automatically. Marking the ROI in plotting UF and AF takes place with the help of the same inbuild atlas, therefore the nerve paths determined this way can be compared objectively. The method used in this case to calculate the fiber length, fiber volume and FA value by the software is based on the same algorithm, therefore the data analysis is objective and repeatable. Moreover, by performing analyzes based on the same protocols, it is possible to eliminate system errors. In the presented case report, the patient's clinical condition was compared based on functional tests (FAST and BNT Tests), taking into account changes in the morphology and anatomy of UF and AF.
We hope that the introduced changes will contribute to a significant improvement in the quality of the article and will make it eligible for publication.
Reviewer 4 Report
See added document

Author Response
Response to review nr 4 round 2
Thank you very much for the professional review and valuable comments on the article. As suggested by the reviewer, the following changes were made:
Introduction:
1) As suggested, the sentence "In this case, perhaps the use of tDCS could contribute to a faster recovery in the young person." has been removed from the article.
Material and Methods:
- Regarding the scales GCS and NIHSS. We strongly agree that when assessing the patient on the NIHSS scale in terms of answering the questions for 2 points (Questions 2 points (patient didn’t correctly answer either question)” it is difficult to evaluate the patient with 15 points on the GCS scale. For this patient, the GCS was routinely used on emergency department admission by a paramedic. This assessment is the standard procedure at the admission to the hospital for each patient. Then, while within the neurology department, it was never assessed on the GCS scale due to the fact that it is not an appropriate scale to assess a patient with aphasia. The condition of the patient in this case is much more adequately reflected in the NIHSS scale, which has been described in detail. The paragraph regarding the GCS scale from the article was removed after a very accurate comment by the reviewer.
- Referring to the differences in FA in both hemispheres at baseline and after 3 months follow up. The DSI Studio that was used to determine the tractography performs data analysis automatically. Marking the ROI in plotting UF and AF takes place with the help of the pre-programmed atlas, therefore the nerve paths determined in this way can be compared objectively. The tractor-FA parameter is not an input parameter, but it is one of the results taken from DSI analysis. I agree with the reviewer that if the researcher entered FA manually, the risk of an error and subjectivization of the results would be very high. In this case, the FA value is the result of the tract visualization so the data analysis seems to be objective and definitely repeatable.
- A phrase about therapy after discharge from hospital has been clarified to contain the rehabilitation guidelines regarding speech: The patient is advised to continue further speech neurological therapy after discharge from the hospital ward provided by trained and qualified speech and language therapists, included tasks devoted to naming, comprehension, and increasing verbal output.
Results :
- The paragraph on pharmacotherapy has been moved as suggested by the reviewer
- Title of paragraph has been changed to: “Outcome at Follow-Up”
- The number of figures has been changed
- Figure 3A is shown from a different perspective (a different angle of the figure has been chosen) to better visualize the AF.
Discussion:
- The part of the description of the AF has been moved to introduction as suggested
2) The paragraph regarding UF has been added to the discussion: “One of the weakness of our study that we didn’t attempt to determine psychological assessment of our patient in terms of empathy and emotional disorders. It was mainly related to the patient's aphasia. Oishi et al. in their study showed a relationship between reduced volume in the uncinate fasciculus and errors on empathy tasks.[38] Previous studies have also proved a relationship between reduced volume of UF and errors on empathy tasks in some neurological disorder. Results confirm that acute damage to the right UF can disrupt performance on a task of emotional empathy. [38,39]. That is reason the psychological assessment of stroke patients is so important in the future studies, taking into account the aspects of emotions and empathy with correlation with the anatomy of UF”
3) Sentence: “It seems that in the case of the patient described in this study, the use of tDCS together with standard techniques of neurologopedic rehabilitation contributed to the recovery of language function.” has been removed.
4) As suggested moval cortex has been changed to motor cortex
5) Sentence “However, successful reperfusion therapy does not end with treatment.” has been changed.
6) The conclusions were corrected as suggested :” This case report demonstrated DTI with tractography as MRI technique used for detect the microstructural changes and difference in the anatomy and morphology of fiber tracts in patient after ischemic stroke which corresponded with improvement in the patient’s clinical functioning. In the described case, the correlation between the patient's clinical improvement and neurorehabilitation enhanced by tDCS cannot be clearly defined. Based on the description of one case, no objective conclusions can be drawn regarding the use of post-stroke tractography and tDCS. The description of this case prompts research on a larger group of patients after ischemic stroke.”
We hope that the introduced changes will contribute to a significant improvement in the quality of the article and will make it eligible for publication.